# Current Status of Neoadjuvant Treatment Before Surgery in High-Risk Localized Prostate Cancer

**DOI:** 10.3390/cancers17010099

**Published:** 2024-12-31

**Authors:** Juan Gómez Rivas, Luis Enrique Ortega Polledo, Irene De La Parra Sánchez, Beatriz Gutiérrez Hidalgo, Javier Martín Monterrubio, María Jesús Marugán Álvarez, Bhaskar K. Somani, Dmitry Enikeev, Javier Puente Vázquez, Noelia Sanmamed Salgado, María Isabel Galante Romo, Jesús Moreno Sierra

**Affiliations:** 1Urology Department, Hospital Clínico San Carlos, 28040 Madrid, Spain; jgrivas@salud.madrid.org (J.G.R.);; 2Health Research Institute, Hospital Clínico San Carlos, 28040 Madrid, Spain; 3Urology, Surgery Department, Universidad Complutense de Madrid, 28040 Madrid, Spain; 4Department of Urology, University Hospital Southampton NHS Foundation Trust, Southampton SO16 6YD, UK; 5Urology Department, Vienna Medical University, 1090 Wien, Austria; 6Institute for Urology and Reproductive Health, Sechenov University, 119435 Moscow, Russia; 7Medicine Dean and Assoc. Deans, Tel Aviv University, Tel Aviv-Yafo 6997801, Israel; 8Urology Department, Rabin Medical Center, Petach Tikva 4941492, Israel; 9Medical Oncology Department, Hospital Clínico San Carlos, 28040 Madrid, Spain; 10Radiation Oncology Department, Hospital Clínico San Carlos, 28040 Madrid, Spain

**Keywords:** prostate cancer, high risk, neoadjuvant, surgery, radical prostatectomy, ADT, chemotherapy, ARTA, lutetium

## Abstract

The likelihood of a biochemical recurrence, metastatic progression and cancer-related mortality after the initial treatment in patients diagnosed with localized high-risk (HR) prostate cancer (PCa) is higher when compared with patients with low (LR) or intermediate-risk (IR) disease. The use of conventional androgen deprivation therapy (ADT) as a neoadjuvant therapy prior to radical prostatectomy (RP) has already been explored by several studies, showing a reduction in positive surgical margins and tumor volume but failing to demonstrate any survival advantages. However, novel agents such as androgen receptor-targeted agents (ARTAs) have exhibited significant survival benefits in both metastatic and non-metastatic castration-resistant PCa, as well as metastatic hormone-sensitive PCa. The significance of the pathological response following neoadjuvant hormonal therapy remains uncertain. In this narrative review, we cover the current status of neoadjuvant therapies before surgery in high-risk localized PCa.

## 1. Introduction

Patients diagnosed with localized high-risk (HR) prostate cancer (PCa) face an elevated likelihood of a biochemical recurrence, metastatic progression and cancer-related mortality after initial treatment, when compared with patients with low (LR) or intermediate-risk (IR) disease [1,2]. The exploration of neoadjuvant therapies before RP for HR PCa is gaining significant momentum, as highlighted by a range of ongoing clinical trials and recent studies. These trials aim to determine whether preoperative treatment can enhance surgical outcomes, reduce tumor burdens and improve long-term survival rates for patients with localized HR PCa.

Several studies have explored the use of conventional androgen deprivation therapy (ADT) as a neoadjuvant therapy to be used prior to a radical prostatectomy (RP) [3,4,5,6,7,8,9,10,11,12,13,14,15]. These studies have shown a reduction in positive surgical margins and tumor volume but have not demonstrated any survival advantages. However, novel agents such as androgen receptor-targeted agents (ARTAs) have exhibited significant survival benefits in both metastatic and non-metastatic castration-resistant PCa, as well as in metastatic hormone-sensitive PCa [16,17,18]. The significance of the pathological response following neoadjuvant hormonal therapy remains uncertain, primarily due to the challenging nature of assessing prostate tissue after hormonal treatment [19]. Pathological evaluations are hindered by the effect of hormonal therapy on the prostate tissue. To overcome these challenges, longitudinal imaging techniques can be employed to identify early responders. These imaging methods can assess the impact of neoadjuvant hormonal therapy on the local tumor, as well as potential micrometastases.

## 2. Materials and Methods

In this narrative review, we cover the current status of neoadjuvant therapies before surgery in HR localized PCa. We performed a comprehensive English literature search for original and review articles through January–August 2024, using Pubmed, Medline and ClinicalTrials.gov databases, as well as a comprehensive review of The American Urological Association (AUA) guidelines, European Association of Urology (EAU) guidelines, European Society for Medical Oncology (ESMO) clinical practice guidelines and National Comprehensive Cancer Network (NCCN) guidelines. We searched for the following terms: neoadjuvant ADT prostate cancer, neoadjuvant ADT, prostate cancer surgery and neoadjuvant high-risk prostate cancer.

The combination of terms found 1314 related articles; the final number of selected papers for this manuscript was 61, including only prospective studies, systematic reviews and meta-analysis and published or ongoing RCTs (phases I, II or III) that included at least 1 arm of HR localized PCa with neoadjuvant treatment prior to RP (Figure 1).

## 3. Results and Discussion

### 3.1. Current Guideline Recommendations

Currently, there is insufficient evidence to create a consensus on neoadjuvant therapy with ADT in HR PCa. The European Association of Urology (EAU) guidelines do not recommend neoadjuvant therapy with ADT prior to RP as a standard clinical practice, based on a systematic review and meta-analysis conducted in 2021 [2,20]. The American Urological Association (AUA) does not include neoadjuvant ADT among its recommendations. However, it is mentioned as a possible option in the future as part of a multimodal treatment to intensify the management of HR patients with localized disease, although the existing evidence is insufficient [21]. The European Society for Medical Oncology (ESMO) clinical practice guidelines make no mention of neoadjuvant therapies prior to surgery [22]. In contrast, the National Comprehensive Cancer Network (NCCN) guidelines strongly discourage the use of ADT as a neoadjuvant therapy outside of clinical trials due to a lack of evidence regarding survival outcomes [23].

### 3.2. Trials on Neoadjuvant Treatment with ADT (1st Generation Anti-Androgens)

ADT can be achieved either with LHRH agonists, LHRH antagonists, anti-androgen treatments or combining a LHRH agonist with an anti-androgen treatment [3,4]. Since the 1990s, several trials have been conducted to study the effect of ADT prior to RP using different pharmacological schemes. The first studies demonstrated significant changes in pathological findings, such as a reduction in pathological stage and a reduction in positive surgical margins or seminal vesicle involvement. However, these studies do not report on medium or long-term oncological outcomes. Additionally, these studies included patients with LR and IR PCa, which may have underestimated the results [4,5].

During the first decade of the 21st century, a new series of trials was conducted to evaluate the duration of neoadjuvant therapies, comparing different strategies (3-month vs. 6-month and 3-month vs. 8-month). These studies also showed significant differences in pathological findings, especially in those undergoing longer ADT schemes. Nonetheless, despite some of these studies evaluating overall survival or disease-free survival (defined by clinical or biochemical progression), they did not find significant differences among the groups [5] (Table 1). Retrospective studies and non-randomized studies specifically targeting HR PCa patients offer similar results on local disease control and local disease control with a slight increase in morbidity, due to ADT-related toxicity (mainly hot flushes and gynecomastia) [5].

### 3.3. Trials on Neoadjuvant Chemotherapy

ADT before surgery in men with localized PCa has shown a histological benefit but no significant improvements in their biochemical progression-free survival [6,11,12,13,14,15,16,17,18,19,20,21,22,23,24,25,26]. In contrast, treatment with Docetaxel has shown an increase in median overall survival in patients with mCRPC [27,28]. Combining both concepts (neoadjuvant therapy in localized PCa and chemotherapy treatment), the CALGB (Alliance) group conducted a study to assess the effect of a neoadjuvant treatment with Docetaxel prior to surgery in patients with high-risk localized PCa [29] by randomizing 788 patients with clinically localized, HR PCa into 2 groups: neoadjuvant chemohormonal therapy (CHT) (391 patients, neoadjuvant arm: Docetaxel with ADT (Leuprolide or Goserelin)) or RP alone (397 patients, surgery arm), with a median follow-up of 6.1 years. The three-year biochemical-PFS (BPFS) (0.89 vs. 0.84, CI 95% for the difference, −0.01 to 0.16), 5-year BPFS (0.81 vs. 0.74, CI 95% for the difference, −0.01 to 0.11; *p* = 0.11) and PCa Cancer-Specific Mortality (CSM) (HR 0.69; 95% CI, 0.32 to 1.07) showed no differences between the treatment groups. However, the overall BPFS (HR 0.69; 95% CI, 0.48 to 0.99), MFS (HR 0.70; 95% CI, 0.32 to 0.95) and OS (HR 0.61; 95% CI, 0.40 to 0.94) were all superior in the neoadjuvant arm.

Previously, several phase I and phase II studies showed that neoadjuvant chemotherapy with Docetaxel prior to RP was well tolerated and led to a reduction in clinical staging [30,31,32]. To assess whether neoadjuvant treatments with Docetaxel and ADT in patients with HR localized PCa improved outcomes compared to those treated exclusively with surgery, a multicenter, randomized phase III clinical trial was designed and conducted. The eligibility criteria were stage cT1–3a PCa (digital rectal examination), a PSA <100 ng/mL, a negative bone scan and no distant disease on abdominal–pelvic computerized tomography (CT) or magnetic resonance imaging (MRI). The definition of HR was determined using the Kattan 1998 nomogram [33], considering patients at a HR to be those with a <60% probability of being biochemical progression-free in the first 5 years. Later, all patients with a Gleason score of 8–10 were also included.

The patients were randomized into two arms: surgery alone or a neoadjuvant therapy with Docetaxel followed by RP. Patients in the neoadjuvant arm received Docetaxel every 3 weeks for 6 cycles, with prior oral dexamethasone and ADT for 18–24 weeks. The primary endpoint was established as 3-year biochemical progression-free survival. The study results demonstrated the adequate tolerability of this neoadjuvant therapy, but do not support the routine use of neoadjuvant chemotherapy in this population due to the lack of significant improvements in 3-year biochemical progression-free survival. These data might be affected by the high percentage of patients (48%) who received early salvage radiotherapy with or without ADT before reaching the primary endpoint (biochemical failure) at 3 years. However, the authors remain confident that in the long term, a more relevant impact will be observed in both primary and secondary endpoints: metastasis-free survival, cancer-specific survival and overall survival.

### 3.4. Trials on Neoadjuvant Treatment with ADT and ARTAs

Since the advent of ARTAs, the management of HR PCa, for both locally advanced and metastatic stages of the disease, has changed. However, there is still a lack of evidence to recommend the use of these drugs as a neoadjuvant therapy before RP [2,3,4,5,6,7,8,9,10]. Currently, the results of some completed phase I, II single-arm studies and randomized controlled trials (RCT) are published, as well as ongoing trials evaluating different strategies both for ADT and ARTAs in a neoadjuvant setting (Table 2).

Bastos et al. [34] conducted a phase II RCT with two arms, including ADT with Abiraterone vs. ADT with Abiraterone + Apalutamide, but their results regarding the oncological outcomes are yet to be published.

Taplin et al. [35,36,37], have conducted two phase II RCTs including IR and HR localized PCa. They compared Enzalutamide + Abiraterone + Prednisone + ADT (ELAP) to Enzalutamide + ADT (EL) (NCT02268175) [34], while also comparing Apalutamide + Abiraterone + Prednisone + Leuprolide (AAPL) with Abiraterone + Prednisone + Leuprolide (APL) (NCT02903368) [35,36]. In the first trial, they recruited 75 patients (50 ELAP, 25 EL), and they reported a favorable pathologic complete response (pCR) with a trend toward improved pathological outcomes with ELAP. In the second study, they recruited 118 patients from four different institutions. AAPL presented similar results in pCR and minimal residual disease (MRD) rates compared to the APL scheme. Their results showed that the addition of Apalutamide did not improve pathologic outcomes.

Fleshner et al. [38] led a multicenter, open-label, phase II RCT in patients with D’Amico HR localized PCa treated with the neoadjuvant Abiraterone + Leuprolide with or without Cabazitaxel [37]. The primary outcome was to determine the pCR or MRD rates. The secondary outcomes included surgical margins, lymph node invasion (LNI), pathologic stage, 12-month biochemical relapse-free survival (BRFS) rates and safety profile. They enrolled 70 patients (Cabazitaxel arm: 37; No Cabazitaxel arm: 33). The pCR/MRD rates were similar in both arms. In contrast, the patients with pCR/MRD had a superior 12-month BRFS (96% vs. 62%, *p* = 0.03). They concluded that the neoadjuvant addition of Cabazitaxel to Abiraterone + Leuprolide prior to RP did not improve the pCR nor MRD in HR localized PCa.

Schweizer et al. [39] conducted an open-label, single-arm phase II trial with HR or very high-risk (VHR) PCa treated with Abiraterone + Apalutamide + Degarelix + Leuprolide + Indomethacin for 12 weeks prior to RP [38]. The primary objective was to determine the pCR rate. The secondary objectives included the minimal residual burden (MRB) and the elucidation of molecular features of resistance. The study recruited 20 patients for its analysis, showing a 5% pCR rate and 30% MRB rate, for an overall pathologic response rate of 35%. Additionally, 90% of the patients were at a ypT3 stage at RP and 35% were N+. They concluded that indomethacin did not add significant benefit to the treatment and that their results contribute to existing literature that suggests that neoadjuvant therapies may have favorable clinical and pathological outcomes.

Lee et al. [40,41] conducted an open-label, single-site phase II trial with IR and HR non-metastatic PCa, using Apalutamide as a neoadjuvant monotherapy for 12 weeks prior to RP. They included 30 patients (20 HR patients), but they did not report pCR in this trial. They observed that biochemical non-responders had a reduced AR activity in pre-treatment biopsies and a reduced androgen response, as well as significantly upregulated innate immune-related pathways like allograft rejection, inflammatory responses and the complement cascade (*p* < 0.05). The results of the self-reported Health-Related Quality of Life outcomes have already been published; the overall outcomes were maintained, but fatigue and erectile dysfunction (ED) were observed [40,41].

The interim analysis for a phase II RCT on neoadjuvant Apalutamide/Abiraterone acetate with Prednisone and the feasibility of performing a nerve-sparing radical prostatectomy in men with HR PCa (NCT02949284) [42] has been published and the early results indicate that receiving neoadjuvant therapy before RP resulted in a reduction in tumor volume and the trends support an increased potency preservation without an adverse effect on positive surgical margin rates.

The ARNEO trial is a double-blind, placebo-controlled trial that evaluates Degarelix +/− Apalutamide as a neoadjuvant therapy before RP in HR PCa [43]. No statistically significant differences were observed in the BCR between the Apalutamide arm and the placebo arm. The primary study endpoint of MRD was not associated with improved BRF, but BFRS was improved in patients with organ-confined disease (ypT2) compared to ypT3-T4. Patients treated with neoadjuvant Degarelix + Apalutamide had improved metastatic disease-free survival compared to Degarelix alone and RP alone (standard of care cohort).

Finally, the PROTEUS trial (NCT03767244) [44], is a large phase III study aiming to determine if Apalutamide + ADT before and after RP is better than placebo + ADT in HR localize PCa or locally advanced PCa, comparing pCR rate and metastasis-free survival.

While these trials represent a significant step forward in the quest to improve outcomes for HR PCa, it is crucial to note that neoadjuvant approaches before prostatectomy are not yet considered standard care.

A summary of ongoing trials on neoadjuvant ADT and ARTAs are included in Table 3 and Table 4.

**Table 2 cancers-17-00099-t002:** Completed trials on neoadjuvant ADT and ARTAs. pCR: Pathological Complete Response, MRB: Minimal Residual Burden; MRD: Minimal Residual Disease; RCB: Residual Cancer Burden; OS: overall survival; BPFS: Biochemical progression-free survival.

Author and RCT	Phase	Inclusion Criteria	Only High-Risk Localized PCa	Neoadjuvant Therapy (Drugs)	Results
Bastos et al. [34] NCT02789878	II Randomized Single masking	T3 (DRE) or Gleason ≥ 8 or PSA > 20	Yes	Goserelin + Abiraterone vs. Goserelin + Abiraterone + Apalutamide	NYR
Taplin et al. [35] NCT02268175	II Randomized Open-label	Intermediate Risk: Gleason 4 + 3 High-Risk: Gleason 8–10 or PSA > 20 or cT3 (MRI)	No	Enzalutamide + Abiraterone + Prednisone + Leuprolide (ELAP) vs. Enzalutamide + Leuprolide (EL)	pCR ELAP: 30% EL: 16% (*p* = 0.26)
Taplin et al. [36,37] NCT02903368	II, Randomized Open-label	Gleason ≥ 7 (4 + 3) Or Gleason 7 (3 + 4) and 1 of the following: - PSA > 20 - ≥cT3 (MRI)	Yes	Arm 1A: Abiraterone + Apalutamide + Leuprolide + Prednisone (AAPL) vs. Arm 1B: Abiraterone + Prednisone + Leuprolide + (APL) Arm 2A: Abiraterone + Apalutamide + Leuprolide + Prednisone (AAPL) 12 months vs. Arm 2B: observation	Part 1: pCR 1A: 22% 1B: 20% (*p* = 0.4) Part 2: BPFS: 2A: 81% 2B: 72% HR: 0.81, 90% CI, 0.43–1.49)
Fleshner et al. [38] NCT02543255	II Randomized Open-label	cT2c-cT3 (DRE+/−imaging) or Gleason ≥ 8 or PSA > 20	Yes	Arm A: Abiraterone + Prednisone + Leuprolide + Cabazitaxel vs. Arm B: Abiraterone + Prednisone + Leuprolide	pCR/MRB Arm A: 43.2% Arm B: 45.5% (*p* = 0.85) pCR Arm A: 5.4% Arm B: 9.1% (*p* = 0.66)
Schweizer et al. [39] NCT02849990	II Single-arm Open-label	High-risk or very high-risk (NCCN Guidelines)	No	Abiraterone + Apalutamide + Degarelix + Leuprolide + Indomethacin	pCR 5% MRD 30% PSA relapse: 14% of pCR group/46% of non-pCR group.
Lee et al. [40,41] NCT03124433	II Single-arm Open-label	D’Amico Intermediate Risk and High-Risk non-metastatic	No	Apalutamide	pCR: 0% Median Cancer Burden Reduction: 41.7%.
Devos et al. [43] NCT03080116	II Randomized Open-label	Intermediate or High-risk PCa	No	Arm 1: Degarelix + Apalutamide vs. Arm 2: Degarelix + Placebo	MRD: Arm 1: 38% Arm 2: 91% CI 95% (*p* = 0.002) PTEN loss: less MRD (11% vs. 43%) (*p* = 0.002). Higher RCB (1.6 vs. 0.4 cm^3^) (*p* = 0.0001)

### 3.5. Trials on Neoadjuvant Treatment with Lu-177

Lutetium-177-PSMA is a radioligand that selectively binds to cancer cells expressing PSMA, inducing cellular damage through the generation of free radicals [55]. There is evidence of its benefits to overall survival in patients with metastatic castration-resistant PCa [56].

Clinical trials like LUNAR [57] have explored its utility in oligorecurrent PCa. Based on the evidence from different therapies already used in other stages of the disease, such as Lu-177, the idea of utilizing them in a neoadjuvant setting for localized PCa has arisen. Golan et al. [58] published the safety profile of Lu-177 at this stage, with a single-arm feasibility study, including patients with localized HR PCa or pelvic lymph node involvement with high PSMA expression in the prostate on previously conducted 68Ga-PSMA PET-CT. The patients received LuPSMA (two or three doses) intravenously at 2-week intervals, and surgery (robot-assisted RP (RARP) with ilio-obturator lymphadenectomy (LND)) was performed 4 weeks after the last dose of LuPSMA. Of the 14 included patients, RARP with LND was performed in 13 cases, all without intraoperative incidents. The PSA decreased by 17% after two doses of LuPSMA and 34% after three doses, although the impact of this decrease is still unknown. Immediate postoperative complications did not increase, and at 3 months post-surgery, 12 patients (92%) required one pad or fewer. In other clinical trials, a higher toxicity associated with LuPSMA was reported, likely due to the higher number of doses used in patients with metastatic PCa and extensive bone involvement. Up to 10% of these patients suffered bone marrow lesions, depicting a different and non-comparable scenario regarding toxicities [56,57,58,59].

The study concluded that LuPSMA administration as a neoadjuvant treatment, followed by RARP, appears to be safe from a surgical perspective. While the oncological results are still pending, post-surgical continence does not seem to be affected by LuPSMA. The study’s limitations are evident due to the small number of patients included. However, by demonstrating the safety of LuPSMA prior to surgery, there is now space for broader clinical trials.

LuTectomy is a prospective phase I/II study of Lutetium-177-PSMA-617 as a neoadjuvant therapy prior to RP [60], for HR PCa (ISUP grade group 3–5, PSA > 20 ng/mL, ≥cT2, N1), M0 and high uptake (SUVmax > 20) on PSMA-PET-CT. It included 20 patients, who received a 5 GBq dose of ^177^Lu-PSMA-617 (Cohort A: 10 patients, Cohort B: 10 patients), followed six weeks later by RP + pelvic LND. The primary outcome was dosimetry (tumor radiation absorbed dose (TRAD)).

The TRAD after cycle 1 for all lesions was 35.5 Gy (Interquartile Range (IQR) 19.5–50.1), with a 19.6 Gy (IQR 11.3–48.4) delivered to the prostate. Five patients received radiation to their lymph nodes. Nine patients (45%) achieved a PSA decline of >50%. The AEs related to ^177^Lu-PSMA-617 were grade 1 fatigue (40%), nausea (35%), dry mouth (30%) and thrombocytopenia (20%). No grade 3/4 toxicities or Clavien 3–5 complications occurred.

The final results of the trial showed that ^177^Lu-PSMA-617 before RP is safe and effective as a neoadjuvant therapy, and its dose delivery appears to be high and targeted, but variable. The biochemical, imaging and pathological responses were encouraging.

### 3.6. Trials on Neoadjuvant Treatments Based on Baseline Genomic Alterations

There are several trials evaluating neoadjuvant treatments based on different genomic alterations. Non-hormonal treatments are being investigated as immunotherapies that can improve anti-tumor immune responses and help to control distant micrometastasis [61].

Recently, B7-H3 was explored as a target for therapy development in PCa as it is highly expressed in this type of cancer. B7-H3 comes from a family of coregulatory molecules, in which PD-L1 and PD-L2 are included. Enoblituzumab, a humanized B7-H3 targeting antibody, was tested in a single-arm study perform by Shenderov et al. [61]. A total of 32 patients with HR localized PCa were treated over 6 weeks with Enoblituzumab prior to RP. This treatment was well tolerated (12% of cases had adverse effects at a grade of 3, with no grade 4 cases) and 66% of patients achieved an undetectable PSA one year after RP.

The genomic umbrella neoadjuvant study (GUNS) is a multi-arm study assessing targeted therapies in patients with selected genomic alterations with HR PCa [62]. Initially, they received 8 weeks of treatment with LHRH agonists and Apalutamide. After that, different target therapies in different subprotocols were added. These subprotocols were as follows: Abiraterone/Prednisone for no targetable actionable aberration; presence of TMPRSS2-ERG fusion, CHD1 loss or SPOP mutations: (~50% expected prevalence in study population), Docetaxel for the loss of tumor suppressor genes—PTEN, TP53 or TB loss (~40%, bad prognosis), PARPi (Niraparib) for DNA damage response alterations (e.g., BRCA1/2, ATM, FANCONI, CDK12) in 6–8% and PD-L1 (Atezolizumab) for hypermutations, microsatellite instability (MSI), Lynch syndrome or CDK12 of less than 5%. The primary endpoint of the study was pCR.

Another ongoing study is the NePtune trial [63], a phase II single-arm study assessing the role of neoadjuvant PARP inhibition with Olaparib followed by RP in patients with HR PCa that present with BRCA 1/2 gene alterations. The treatment duration of Olaparib prior to RP is 6 months. The primary endpoints are as follows: pCR and MRD.

### 3.7. Molecular and Pathological Response Markers in Neoadjuvant Therapy

There is increasing interest in knowing the molecular markers that might help with predicting treatment responses, given the heterogeneity of HR PCa. While neoadjuvant therapy may show benefit in some patients treated with RP, inducing pCR or MRD, there is a growing interest in the underlying biology and molecular markers of treatment responses, which remain unclear.

Although anatomopathological response is a marker for estimating long-term survival in several tumor types, it remains unclear regarding PCa. Mckay et al. [36] performed a multicenter phase II study to assess the impact of pathological responses on recurrence rates in patients treated with intense neoadjuvant treatments prior to RP. They found that PTEN-loss and ERG + expression were associated with larger residual tumors and a higher residual cancer burden (RCB). The presence of intraductal carcinoma was found to be another marker of a higher T stage, residual tumor size or residual cancer.

In the ARNEO study [43], the investigators introduced the role of PET-PSMA as biomarker to assess responses to neoadjuvant hormonal treatment in patients treated with Degarelix with or without Apalutamide prior to RP. They concluded that PET-PSMA estimated tumor volumes, finding that the PSMA SUVmax after neoadjuvant therapy was lower in patients with minimum residual disease. Another crucial value was the PSA, showing that a Nadir after therapy of < 0.3 ng/mL was associated with a lower RCB. Like in the other studies mentioned before, a PTEN loss was associated with a higher RCB.

Tewari et al. [64] analyzed molecular features in patients treated with intense neoadjuvant therapy prior to RP (with Enzalutamide, ADT, Abiraterone and Prednisone or ADT with Apalutamide and Abiraterone), comparing patients with exceptional responses (ERs) and the ones that had non-responses (NRs). An ER was understood as complete response or with <5 mm residual tumor after RP. No differences were seen between groups in the tumor mutational burden or in the proportion of genome alterations. They found that recurrent mutations in SPOP were seen exclusively in the ER group, and that TP53 mutations, PTEN loss, PHLPP1 loss and androgen receptor deletions were exclusively in the NR group. More upregulated genes were identified in ER (136 genes) compared with the NR group (21 genes). The presence of TMPRSS2-ERG fusions were more frequent in the NR group of their cohort, and an increase in TGF-beta signaling was seen in the NR group.

These findings may help while accurately choosing a tailor-made neoadjuvant therapy. More trials will be needed to better characterize the molecular alterations of these tumors and help to predict the pathology response after a neoadjuvant treatment. An ideal marker should help us identify non-responders prior to RP in order to avoid any delay in a definitive treatment, but there is little evidence regarding the clinical impact and molecular determinants, and the impact of delaying definitive treatment in this setting.

Ongoing and future studies focusing on neoadjuvant therapies should incorporate translational endpoints. This approach will aid in identifying early resistance mechanisms and developing innovative biomarkers. Ultimately, these advancements will enable personalized neoadjuvant hormonal treatments for patients diagnosed with high-risk PCa.

## 4. Conclusions

Currently, no phase III data supports the use of systemic neoadjuvant therapies before RP. Phase II studies show promising data for ADT with second-generation agents, including a favorable pCR and MRD, along with PSA relapse and salvage therapy rates. Ongoing and future studies focusing on neoadjuvant therapies should incorporate translational endpoints, helping to identify early resistance mechanisms and develop innovative biomarkers. Eventually, this will enable us to tailor neoadjuvant hormonal treatments for patients diagnosed with HR PCa.

## Figures and Tables

**Figure 1 cancers-17-00099-f001:**
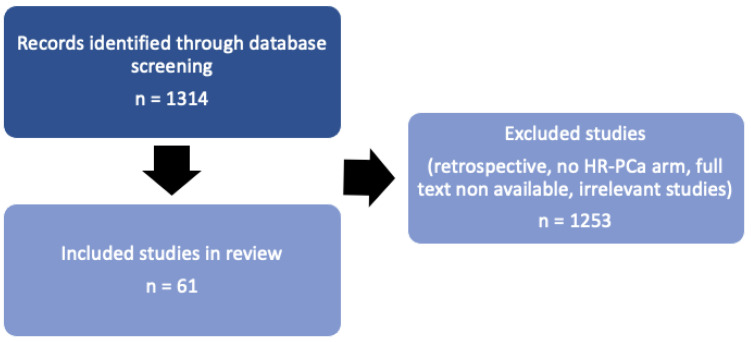
Screening and inclusion diagram. HR PCa: high-risk prostate cancer.

**Table 1 cancers-17-00099-t001:** Trials on neoadjuvant ADT (1st generation anti-androgens). OS: overall survival; PFS: progression-free survival; NR: not reported.

Author	n	Clinical Stage	Neoadjuvant Therapy (NADT) (Months)	Positive Margins (%) NADT vs. Only Surgery	Seminal Invasion (%) NADT vs. Only Surgery	pN+ (%) NADT vs. Only Surgery	OS (%) NADT vs. Only Surgery	PFS (%) NADT vs. Only Surgery	Follow Up (Years)
Dalkin et al. [6] 1996	56	T1c–T2b	Goserelin (3)	18% vs 14%	-	-	NR	NR	NR
Goldenberg et al. [7] 1996	213	T1b–T2c	Cyproterone (3)	28% vs. 65%	28% vs. 14%	7% vs. 3%	NR	NR	NR
Labrie et al. [8] 1997	161	T2–T3	Leuprolide + Flutamide (3)	7.8% vs. 33.8%	NR	NR	NR	NR	NR
Fair et al. [9] 1999	140	T1–T2	Goserelin + Flutamide (3)	19% vs. 37%	NR	NR	NR	NR	2.9
Schulman et al. [10] 2000	402	T2–T3	Goserelin + Flutamide (3)	13% vs. 37%	NR	NR	96% vs. 96%	74% vs. 67%	4
Aus et al. [11] 2002	126	T1b–T3a	Triptorelin + Cyproterone (3)	23.6% vs. 45.5%	NR	NR	83% vs. 86%	49.8% vs. 51.5%	6.85
Soloway et al. [12] 2002	303	T2b	Leuprolide + Flutamide (3)	18% vs. 48%	15% vs. 22%	6% vs. 6%	NR	64.8% vs. 67.6%	5
Selli et al. [13] 2002	431	T2–T3	Goserelin + Bicalutamide (3 or 6 months)	28% (3 months)/23% (6 months) vs. 53%	11% (3 months)/11% (6 months) vs. 11%	8% (3 months)/4% (6 months) vs. 12%	NR	NR	NR
Klotz et al. [14] 2003	213	T1b–T2	Cyproterone (3)	28% vs. 65%	NR	NR	93% vs. 95%	60.2% vs. 68.2%	6
Prezioso et al. [15] 2004	183	T1a–T2b	Leuprolide + Cyproterone (3)	39% vs. 60%	NR	3% vs. 11%	NR	NR	NR
Yee et al. [24] 2010	148	T1b–T3	Goserelin + Flutamide (3)	19% vs. 38%	4% vs. 6%	1% vs. 3%	86% vs. 92%	80% vs. 78%	8

**Table 3 cancers-17-00099-t003:** Ongoing randomized trials on neoadjuvant ADT and ARTAs. ADT: Androgen Deprivation Therapy; ARN-509 (Androgen Receptor Antagonist ARN-509).

Author and RCT	Phase	Inclusion Criteria	Only High-Risk Localized PCa	Neoadjuvant Therapy (Drugs)	Status
Ghodoussipour et al. [42] NCT02949284	II Randomized Open-label	≤T3 and Gleason ≥ 8 (4 + 3) and PSA >20 and >1 core	Yes	ARN-509 vs. Abiraterone + Prednisone + ARN-509 + GnRH Analogue vs. None	Recruiting
Janssen Research & Development, LLC [44] NCT03767244	III Randomized Double blind	High-risk localized or locally advanced	Yes	Apalutamide + ADT vs. Placebo + ADT	Active, not recruiting
Szmulewitz et al. [45] NCT05726292	II Randomized Double-blind	≥cT3 or ISUP ≥ 4 or PSA >20	Yes	Enzalutamide + Relacorilant + ADT vs. Placebo + ADT	Recruiting
Ravi et al. [46] NCT05617885	II Randomized Open-label	Gleason ≥ 8 Or Gleason 7 (4 + 3) and 1 of the following: - PSA > 20 - ≥cT3 - EPE on - biopsy	Yes	Abemaciclib + Darolutamide + ADT vs. Darolutamide + ADT	Active, not recruiting
Reiter et al. [47] NCT01990196	II Randomized Open-label	≥cT3a or Gleason ≥ 7 (4 + 3) or any Gleason 5 or PSA > 20	Yes	Degarelix + Enzalutamide vs. Degarelix + Enzalutamide + MEK-inhibition (Trametinib) vs. Degarelix + Enzalutamide + SRC-inhibition (Dasatinib)	Active, not recruiting

**Table 4 cancers-17-00099-t004:** Ongoing single-arm and non-randomized trials on neoadjuvant ADT and ARTAs.

Author and RCT	Phase	Inclusion Criteria	Only High-Risk Localized PCa	Neoadjuvant Therapy (Drugs)	Status
Zeng et al. [48] NCT04736108	II Single-arm Non-randomized	Intraductal carcinoma + ≥cT3a or Gleason ≥ 8 or PSA > 20	Yes	Abiraterone + Goserelin + Prednisone	Not yet recruiting
Zhang et al. [49] NCT04997252	II Single-arm Non-randomized	High-risk localized PCa or Oligometastatic PCa	No	Apalutamide + LHRH agonist	Recruiting
Hongqian et al. [50] NCT04356430	II Single-arm Non-randomized	≥cT3 (PSMA PET/CT or mpMRI) or Gleason ≥ 8 or PSA > 20	Yes	Abiraterone + Goserelin + Prednisone	Unknown
Hongqian et al. [51] NCT05249712	II Single-arm Non-randomized	≥cT3 (PSMA PET/CT or mpMRI) or Gleason ≥ 8 or PSA > 20 or N1	No	Darolutamide + ADT	Recruiting
Hongqian et al. [52] NCT05406999	II Non-randomized Open-label	≥cT3 (PSMA PET/CT or mpMRI) or Gleason ≥ 8 or PSA > 20 or N1	No	ADT alone vs. ADT + Abiraterone vs. ADT + Enzalutamide vs. ADT + Apalutamide vs. ADT + Darolutamide vs. ADT + Rizvilutamide vs. ADT + PARP inhibitor + Abiraterone vs. ADT + PARP inhibitor	Recruiting
Kopp et al. [53] NCT05593497	II Single-arm Open-label	Gleason > 7 (>ISUP-3) or PSA > 20 (<T3 or Gleason < 7 must have 5-year PF probability ≤50% estimated by MSKCC pre-radical prostatectomy nomogram) + ≤10% PTEN staining	Yes	Leuprolide + Abiraterone + Capivasertib	Recruiting
Zhu et al. [54] NCT05223582	II Single-arm Open-label	High-risk or very high-risk (NCCN Guidelines)	No	Abiraterone + Fluzoparib + Prednisone + ADT	Active, not recruiting

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
