# Peer review of "Current Status of Neoadjuvant Treatment Before Surgery in High-Risk Localized Prostate Cancer"

_cancers, 2024, doi:10.3390/cancers17010099_

Round 1
Reviewer 1 Report (Previous Reviewer 1)
Comments and Suggestions for Authors
Revised paper has addressed previous concerns.
Author Response
Comment 1: Please provide the institutional emails/CV, not gmail.com/yahoo.com/hotmail.com.
Response 1: Thank you for pointing this out. We added the institutional emails to the paper. In addition, we have updated Dr. Enikeev's affiliations.
Comment 2: I would like to inform you that we have compared your manuscript with the published materials and found that a part of your text is showing significant overlap with already published sources. I would like to ask you kindly to reduce the overlap in your manuscript during the revision. Sames as the green highlight below.
Here are some examples how can overlap be reduced:
- Make different sentence order;
- Use different words wherever is possible;
- Add adjectives as many as you can;
- Delete some sentences if not necessary;
- Merge 2 sentences into 1 with the same meaning.
Response 1: Thank you for this correction. We have reshaped the highlighted paragraphs to reduce overlap. They are now highlighted in light blue.
This manuscript is a resubmission of an earlier submission. The following is a list of the peer review reports and author responses from that submission.
Round 1
Reviewer 1 Report
Comments and Suggestions for Authors
The authors seek to summarize literature to date regarding neoadjuvant therapies prior to radical prostatectomy for higher risk prostate cancer.
Overall while the aims of the review are laudable, the work as presented seemed disjointed and presentation of results were hard to follow.
Specific comments/suggestions:
Omit it Table 1 - it is summarized fully within the text.
Table 2 is formatted poorly and is hard to interpret. The experimental and control arms and results for each arm for each study quoted should be clearly presented.
Section 3.3 specify what were the clinical outcomes noted in the alliance trial (PFS, OS) by treatment arm.
Section 3.4 many of the studies quoted are referred to in the text as RCT (randomized controlled trials) but are in fact single arm phase II trials. The authors should organize 3.4 and Table 4 to separate single arm trials and randomized trials so it is easier to interpret the evidence base. Table 4 should include the results of the studies not just a reference to the paper in the results section of the table. Separate studies by completed and in progress.
Section 3.5 should have more details about LuTectomy in particular the pathologic findings of this study. This study should have more emphasis than the Golan study as it accrued more patients and reports all the same endpoints. Also, in LuTectomy some men had a single dose, some had two doses of Lu.
In 3.6 the authors should clarify how men are assigned to therapies on the GUNS trial (i.e. what markers determine what treatment arms)
Sections 3.7 and 3.8 overlap and include non-RP populations/examples, the authors should focus on assessment of response pre-RP and in particular this would be a good place to feature the utility of "window of opportunity" studies where new therapeutic agents are administered pre-surgery in order to assess biologic effect (rather than therapeutic benefit) - LuTectomy is a good example of a WOS. Apart from use in WOS context, how does response assessment prior to RP influence care?
Comments on the Quality of English Language
Acceptable
Author Response
Comment 1: Omit it Table 1 - it is summarized fully within the text.
Response 1: Thank you for pointing this out. Omitting Table 1 leads to a rearrangement of the rest of the tables (i.e. Table 2 is now Table 1 and so on).
Comment 2: Table 2 is formatted poorly and is hard to interpret. The experimental and control arms and results for each arm for each study quoted should be clearly presented.
Response 2: Thank you for suggesting this improvement. Table 2 (now renamed Table 1, see above) is now reformatted. Full names of the different Neoadjuvant ADT schemes are now provided. In each result column it is now specified that the first result is related to the Neoadjuvant ADT arm (experimental), whereas the second result is related to the control arm (only surgery).
Comment 3: Section 3.3 specify what were the clinical outcomes noted in the alliance trial (PFS, OS) by treatment arm.
Response 3: Clinical outcomes have been added to the manuscript.
Comment 4: Section 3.4 many of the studies quoted are referred to in the text as RCT (randomized controlled trials) but are in fact single arm phase II trials. The authors should organize 3.4 and Table 4 to separate single arm trials and randomized trials so it is easier to interpret the evidence base. Table 4 should include the results of the studies not just a reference to the paper in the results section of the table. Separate studies by completed and in progress.
Response 4: Table 3 (now renamed Table 2) and its references have been reorganized, and their results have been added to the table (completed trials). Table 4 has been divided into 2 tables: Table 3 (for ongoing randomized trials) and Table 4 (for ongoing single-arm and non-randomized trials). All references have been adapted to the content of the tables and manuscript.
Comment 5: Section 3.5 should have more details about LuTectomy in particular the pathologic findings of this study. This study should have more emphasis than the Golan study as it accrued more patients and reports all the same endpoints. Also, in LuTectomy some men had a single dose, some had two doses of Lu.
Response 5: Section 3.5 has been modified by adding more details about LuTectomy, including the pathological findings.
Comment 6: In 3.6 the authors should clarify how men are assigned to therapies on the GUNS trial (i.e. what markers determine what treatment arms)
Response 6: A description of the 4 sub-protocol selection has been added to the manuscript.
Comment 7: Sections 3.7 and 3.8 overlap and include non-RP populations/examples, the authors should focus on assessment of response pre-RP and in particular this would be a good place to feature the utility of "window of opportunity" studies where new therapeutic agents are administered pre-surgery in order to assess biologic effect (rather than therapeutic benefit) - LuTectomy is a good example of a WOS. Apart from use in WOS context, how does response assessment prior to RP influence care?
Response 7: Many thanks for this appraisal. We have merged Section 3.7 and 3.8 with a new definition: Molecular and pathological response markers in neoadjuvant therapy. We have deleted 2 references that mentioned non-RP populations (Raipar et al (63) from section 3.7 and De Bono (66) from section 3.8) and the text has been rearranged. If we consider the concept “window of opportunity” in this clinical setting, an ideal marker should help us identify non-responders prior to RP in order to avoid any delay in definitive treatment, but there is little evidence regarding the clinical impact and molecular determinants and the impact of delaying definitive treatment in this setting. We have added this consideration in the manuscript.
Reviewer 2 Report
Comments and Suggestions for Authors
I have reviewed the manuscript titled “CURRENT STATUS OF NEOADJUVANT TREATMENT BEFORE SURGERY IN HIGH-RISK LOCALIZED PROSTATE CANCER.” The authors aim to examine various neoadjuvant therapies available for high-risk prostate cancer. Overall, I find the paper to be well-written with clearly defined objectives. The topic is timely and warrants further exploration, especially given the discrepancies present in the recommendations of different international guidelines. The data presented are comprehensive and well-elaborated. As such, I recommend the manuscript for acceptance, pending minor formatting editing.
1. In Table 2, please correct the heading from “autor” to “author.”
2. In subchapter “3.4. Trials on Neoadjuvant Treatment with ADT and ARTAs”, please ensure proper citations are provided for each described study. For instance, on line 180, the reference for Bastos et al. is missing and should be added. Please rectify similar issues throughout this section.
3. In the annotation for Table 3, please provide the abbreviations for IR (Intermediate Risk) and HR (High Risk) in reference to Taplin et al. for clarity.
Author Response
Comments 1: In Table 2, please correct the heading from “autor” to “author.”
Response 1: Many thanks for pointing out this error. After extensive review, Table 2 has been renamed as Table 1, and this mistake has been fixed.
Comments 2: In subchapter “3.4. Trials on Neoadjuvant Treatment with ADT and ARTAs”, please ensure proper citations are provided for each described study. For instance, on line 180, the reference for Bastos et al. is missing and should be added. Please rectify similar issues throughout this section.
Response 2: Thank you for mentioning this mistake. There was no reference to Bastos et al. in the References Section, therefore it has been added, along with the rest of missing references. All references have been rearranged to pair the manuscript.
Comments 3: In the annotation for Table 3, please provide the abbreviations for IR (Intermediate Risk) and HR (High Risk) in reference to Taplin et al. for clarity.
Response 3: Table 3 has been renamed as Table 1. We have provided the whole reference (Intermediate Risk and High Risk) rather than abbreviation in order to make it cohesive with the rest of the text.